# Improving the quality of maternal and newborn healthcare at the district level: Addressing newborn deaths in Nepal

**Subaru Ikeda**[1], **Akira Shibanuma**[1]*, **Alpha Pokharel**[1,2], **Ram Chandra Silwal**[1], **Masamine Jimba**[1]

1 Community and Global Health, Graduate School of Medicine, the University of Tokyo, Tokyo, Japan,
2 Lawrence S. Bloomberg Faculty of Nursing, University of Toronto, Toronto, Canada

* shibanuma@m.u-tokyo.ac.jp

**Data Availability Statement:** For this study, we received approval for using NSPA and NDHS datasets from the DHS program and downloaded

## Abstract

Maternal and newborn care quality can be measured in three dimensions (Dimensions 1: care provision, 2: care experience, and 3: human and physical resources); however, little is known about which dimensions are associated with newborn and perinatal deaths. We examined the association between care quality and newborn and perinatal deaths in Nepal. This study incorporated secondary data from Nepal Service Provision Assessments (NSPA) 2015 (623 delivery facilities, facility inventory survey; 1,509 women, ANC clients interviews; 1,544 women, ANC observation) and Nepal Demographic and Health Surveys (NDHS) 2016 (5,038 women who reported having given birth in the five years preceding data collection). The outcome variables were newborn and perinatal deaths derived from the NDHS. The exposure variables were district-level maternal and newborn care quality scores calculated from the NSPA data. Covariates were women's sociodemographic, health, and obstetric characteristics. We applied the administrative boundary method to link these two surveys. We conducted binary logistic regression analyses to examine the association between care quality and newborn/perinatal deaths. In Dimension 1, higher mean and maximum quality scores at the district level were associated with a lower number of newborn deaths (mean: odds ratio [OR] = 0.04, 95% confidence interval [CI]: 0.00–0.76; max: OR = 0.09, 95% CI: 0.01–0.58), but not with perinatal deaths. In Dimensions 2 and 3, the quality score was not significantly associated with newborn deaths and perinatal. Enhancing the quality of care provision at its average and highest levels in each district may contribute to the reduction of newborn deaths, but not perinatal death. Health administrators should assess the quality of care at the administrative division level and focus on enhancing both average and maximum care quality of health facilities in each region in the care provision dimension.

## 1. Introduction

During the Millennium Development Goals era, the utilization of facility delivery increased, but it did not necessarily result in improving maternal and child health outcomes [1].

the data from their website (https://dhsprogram.com).

**Funding:** SI was supported by JST SPRING, Grant Number JPMJSP2108 (https://www.jst.go.jp/jisedai/) and Kazuharu Ogura Scholarship for Postgraduate Education (International Nursing) (https://www.nurse.or.jp). AS receives Japan Society for the Promotion of Science KAKENHI Grant Numbers JP21H03160 (https://www.jsps.go.jp). The funders had no role in study design, data collection and analysis, decision to publish, or preparation of the manuscript.

**Competing interests:** The authors have declared that no competing interests exist.

**Abbreviations:** ANC, Antenatal Care; AOR, Adjusted Odds Ratio; BEmOC, Basic Emergency Obstetric Care; CI, Confidence Interval; IQR, Interquartile Rang; LMICs, Low- and Middle-Income Countries; NDHS, Nepal Demographic and Health Surveys; NSPA, Nepal Service Provision Assessments; DHS, Demographic and Health Surveys; OR, Odds Ratio; PCA, Principal Components Analysis; PCMC, Person-Centered Maternity Care; SBA, Skilled Birth Attendants; SD, Standard Deviation; SPA, Service Provision Assessments; WHO, World Health Organization.

According to an analysis of 67 low- and middle-income countries (LMICs), the association between facility delivery and early neonatal mortality was inconsistent [2]. Out of 67 countries that assessed the association, the odds of early neonatal mortality were reduced in 16 countries, but they were increased in 10 countries, and the majority had no significant association [2]. In LMICs, amenable maternal and newborn deaths owing to poor-quality care exceed those of non-utilization of available healthcare. In 2016, an estimated 657,555 newborns and 56,634 mothers died owing to poor-quality care [3]. Ensuring high-quality care will be key to improving maternal and newborn health in LMICs.

According to the World Health Organization (WHO), the quality of care for women and newborns is "the degree to which maternal and newborn health services (for individuals and populations) increase the likelihood of timely, appropriate care for the purpose of achieving desired outcomes that are both consistent with current professional knowledge and take into account the preferences and aspirations of individual women and their families [4]." The quality of maternal and newborn care is a multidimensional concept, including "the quality of care provision" and "the quality of care experience of women, newborns, and their families [4]." The WHO framework for the quality of maternal and newborn care comprises eight domains. These domains are categorized into three dimensions: the provision of care, the experience of care, and cross-cutting issues between these two dimensions. The provision of care dimension has three domains: 1) evidence-based practices for routine care and management of complications, 2) actionable information systems, and 3) functional referral systems [4]. The experience of care dimension also has three domains: 4) effective communication, 5) respect and preservation of dignity, and 6) emotional support. The remaining domains exist as cross-cutting issues: 7) competent, motivated human resources and 8) essential physical resources available [4]. This multidimensional concept is an essential element of the right to health and the path to equity and dignity for women and children [5,6].

To change maternal and newborn care performance, high-quality care at each health facility is necessary but insufficient [1]. For instance, even if a health facility is high-quality, a low-quality referral system could produce a further delay in receiving care and deteriorate health outcomes [7]. Improving the whole health system is required to improve health outcomes; that is, appropriate governance that fits the target areas, proper allocation of health resources, early detection of complications, appropriate diagnosis and emergency care at primary-level facilities, timely referral, adequate inter-facility information sharing, and intensive treatment at secondary- and tertiary-level facilities [1].

Nepal is a lower-middle-income country in which high-quality care is needed to improve maternal and newborn health outcomes. According to a descriptive study that assessed the quality of primary care across 10 LMICs, composite and domain scores for quality of primary care facilities were low, particularly in user experience across 10 countries including Nepal [8]. In a study in Nepal, skilled birth attendants' (SBAs) knowledge and clinical skill score on basic maternal and newborn care were far from the national standard, and only seven percent of them met the minimal delivery volume to maintain competence recommended by the WHO [9]. According to a study conducted in two hospitals in Nepal, all women experienced disrespectful/abusive care, such as no consent, non-dignified care, or non-confidential care [10]. In a qualitative study in a mountainous area in Nepal, women experienced low-quality care, with women's feeling unsafe and uncomfortable at health facilities owing to poor cultural acceptance, absenteeism, negligence of health workers, and corruption [11]. According to the government report, only five percent of health facilities handling delivery met the requirements in a national guideline about basic medical resources [12]. Moreover, in a study that assessed the quality of care by receipt of basic components of antenatal care (ANC) in 2007–2016, an inequality index of the quality of care was the largest in Nepal among 30 low-income countries

[13] (classification of Nepal changed to a lower-middle-income country in 2020 [14]). Especially, vulnerable populations may have disadvantages in the uptake of high-quality care [13]. Ensuring universal access to high-quality basic maternal and newborn care is an urgent issue in Nepal.

To measure the level of maternal and newborn care quality, previous studies have used data on components of essential care, infrastructure, equipment, supplies, and staffing [13, 15]. However, these studies overmeasure inputs and are insufficient to measure the health system as a whole [1]. Although many studies have focused on the coverage of routine maternal and newborn care and the availability of physical resources, few studies have systematically examined multidimensional maternal and newborn care quality, including experience of care [13, 16, 17]. No studies have examined district-level multidimensional care quality to date [18]. Although a study in another South Asian country examined the geographic variation of maternal and newborn care quality at the district level, it also focused on inputs of routine care but not multidimensional care quality [19]. The following two types of nationally representative data are available for maternal and newborn health in many low- and lower-middle-income countries: demographic and health surveys (DHS) and service provision assessments (SPA). SPA covers three dimensions of the care quality in the WHO framework, and DHS covers women's characteristics and health outcomes [20]. Although some studies have investigated family planning and facility readiness by linking DHS and SPA, few studies have investigated multidimensional care quality using both datasets [21]. Few studies examined an association between the care quality and health outcomes [8, 18]. However, multidimensional quality, including women's experience and its association with health outcomes, has not yet been examined [13]. Therefore, we conducted this study to examine the association between district-level maternal and newborn care quality and newborn/perinatal deaths in Nepal.

## 2. Methods

### 2.1. Ethics statement

The Research Ethics Committee of the Graduate School of Medicine, the University of Tokyo, approved this research (no. 2021189NI). The original survey conductors of SPA and DHS obtained ethical approval for the data collection procedures, and they also obtained written informed consent from mothers preceding the surveys.

### 2.2. Data source and samples

In this cross-sectional study, secondary data were obtained from the 2015 Nepal Service Provision Assessments (NSPA) and the 2016 Nepal Demographic and Health Surveys (NDHS). The MEASURE DHS project conducted NSPA and NDHS as nationally representative surveys in Nepal [22, 23]. For this study, we received approval for using NSPA and NDHS datasets from the DHS program and downloaded the data from their website (https://dhsprogram.com).

Data of NSPA 2015 were collected in two phases owing to the earthquake that occurred during the survey: April 20 to 25, 2015, and June 4 to November 5, 2015. In the NSPA, 992 health facilities were selected from 4,719 health facilities in Nepal. The selected facilities included all nonspecialized government hospitals, all private hospitals with 100 or more inpatient beds, and all primary health care centers. In addition, health posts, private hospitals whose beds were between 15 and 99 beds, stand-alone HIV testing and counseling sites, and urban health centers were included as a result of sample selection to set the total number of facilities as approximately 1,000. Of these, 963 health facilities participated in the survey, excluding 29 health facilities that were permanently closed, unreachable, or refused to participate. For ANC client exit interviews in the NSPA, interviewers attempted to interview all

clients who visited the health facility on the day of the interview until it reached a target number. A maximum of 15 women at each facility were recruited out of 4,498 women presented at the surveyed facilities on the day of the survey at 180 ANC facilities. From the dataset of the NSPA, we used facility inventory survey data from 623 delivery facilities, interviews with 1,509 ANC clients, and observations of 1,544 ANC consultations.

NDHS 2016 data were collected between June 19, 2016, and January 31, 2017. In the NDHS, out of 13,089 sampled women, 12,862 women aged 15–49 years completed the interview (response rate: 98.3 percent). We used data from 5,038 women who reported having given birth in the five years preceding data collection.

We included 72 districts of all 75 districts (Nepal has 77 districts after the national structural change in 2015, but the NSPA 2015 and NDHS 2016 were conducted based on the previous structure frame), where both SPA and DHS conducted the survey; Three districts (Manang, Dolpa, and Mugu) were excluded since the data from them were not available.

## 2.3. Data linkage

We applied an administrative boundary method to link the two surveys, using "district" as a geographical identifier [21, 24]. This method does not require GPS data, and DHS clusters are connected with health facilities that have the same geographic identifiers in SPA. In this study, women's characteristics and health outcomes were connected with the representative data of health facilities in the same district [25]. We used this method since we focused on examining the quality of care at the district level. Other methods, for example, connecting DHS clusters with the closest health facilities to their residences, can examine the facility-level quality of care. However, these methods may misclassify women with health facilities when women used unsampled facilities in the survey since NSPA is not a census of health facilities. Then it may lose the characteristics of the quality of that region. Even though it is not a census of health facilities, the administrative boundary method can be used since the health facility data is representative of the region. Moreover, the administrative boundary method could eliminate the mismatch owing to cluster displacement in the DHS sampling procedure and women's preference for health facilities within the district [26] (Fig A in S1 Text).

## 2.4. Variables

The outcome variables were newborn and perinatal deaths, which were measured using the NDHS (Fig B in S1 Text). Newborn deaths are defined as deaths within 28 days after the births of children, and perinatal deaths are defined as fetal deaths that occurred after 28 weeks of gestation or child deaths occurred within seven days after birth [27]. These are binary variables of whether a woman experienced the deaths of her children at least once. In this study, we used the variables "b5" (if a child is alive) and "b6" (days at death measured as the number of days after birth) in BR file in the NDHS dataset to calculate neonatal deaths. Children born and dead within 28 days after birth were counted as neonatal deaths. Children born and dead within seven days after birth were counted as early neonatal deaths. We used the variable "s223a" to calculate stillbirths. If "s223a" is one or greater, they were counted as stillbirth. Perinatal death was a stillbirth or early neonatal death. We included data on newborn and perinatal deaths among all women who had given birth five years prior to the interview. The data did not reflect the cases in which mother-child dyads died since these data were based on interviews with women about their past birth experiences.

The exposure variables were the three dimensions of maternal and newborn care quality scores at the district level, measured based on the NSPA: Dimension 1: provision of care, Dimension 2: experience of care, and Dimension 3: human and physical resources. Based on

prior research, we chose indicators for Dimensions 1 and 3 from the facility inventory dataset and indicators for Dimension 2 from ANC client exit interviews and the ANC observation dataset in the NSPA [4, 17, 28–30]. Indicators in Dimension 1 included Basic Emergency Obstetric Care (BEmOC) signal functions which were related to both women and children, and some indicators were directly related to maternal deaths rather than newborn and perinatal deaths. However, BEmOC is widely introduced into LMICs as a package of care, and prior studies showed its effect on newborn and perinatal deaths [31–33]. Thus, this study included all seven signal functions as indicators of Dimension 1.

We first calculated the facility-level quality scores and then calculated the district-level quality score based on them. At the facility-level analysis, we used an additive indicators method and the principal components analysis (PCA) method [34]. In the additive indicators method, the quality score at each facility was the simple summation of the indicators within each dimension. The sum of the indicators was divided by the number of indicators so that it took between 0 and 1. In the PCA method, we used the principal component score from the first eigenvector of the indicators to obtain quality scores at each health facility. We then used a correlation coefficient to ensure that the scores derived from the different methods did not have a significant effect on the results. We used the score from an additive indicator method because these scores were simpler and understandable [34].

We then calculated district-level quality scores using these facility-level quality scores. We calculated the mean, median, and maximum quality scores for the facilities in each district. We compared the mean and median scores because extreme outliers may affect the mean scores. We also compared the mean and maximum scores to reflect the different perspectives on the district's health facility allocation: having a high-quality average among health facilities in a district and having at least one high-quality health facility in a district. The quality score in each dimension was "Quality Score 1," "Quality Score 2," and "Quality Score 3," respectively.

Quality Score 1, or provision of care, had seven indicators on information system and BEmOC signal functions: 1) partograph use (graphical records specific for labor and delivery), 2) capacity to remove retained products, 3) parenteral oxytocin for hemorrhage in the last three months, 4) parenteral magnesium sulfate for (pre-) eclampsia in the last three months, 5) manual removal of placenta in the last three months, 6) parenteral antibiotics for a maternal infection in the last three months, and 7) neonatal resuscitation in the last three months [29]. BEmOC is a package of care for maternal and neonatal life-threatening complications. The WHO guidelines recommend that all women who give birth should have access to them [35]. Out of BEmOC indicators, 2), 3), and 5) might be more related to women's physical condition than children's. However, they are widely introduced into LMICs as a package of care, and prior studies have shown their effect on newborn deaths and stillbirths [31–33]. Thus, this study included all seven signal functions as indicators of Quality Score 1.

Quality Score 2, or experience of care, had four indicators: 1) women's perception of ANC, 2) women's satisfaction with ANC, 3) the provider asked if a client had any questions, and 4) the provider used visual aids during consultation. We reviewed the SPA questionnaires and extracted indicators related to this dimension based on the WHO framework.

Quality Score 3, or human and physical resources, had five indicators: 1) 24-hour availability of skilled providers, 2) referral resources, 3) electricity, 4) improved water source, and 5) capacity of infection control based on prior research [29]. Indicators were binary except for infection control, women's perception, and satisfaction. The binary indicator has a value of either zero or one. A continuous indicator was transformed such that it ranged from zero to one (Table A in S1 Text).

We obtained data on women's age, ethnicity, educational attainment, residential area, wealth index, intended pregnancy, multipara, involvement in decision-making on health,

multiple pregnancies, tobacco, and the province as covariates based on previous studies [1, 18, 29, 36]. We also used the province in which women resided as a covariate to capture the characteristics of a province that were not measured by the other covariates because the individual data have some degree of similarity within a cluster. We selected these variables which possibly affect either exposure or outcome variables [37, 38].

### 2.5. Analysis

Binary logistic regression models were used to examine the association between care quality and newborn and perinatal deaths. Considering the cluster effects, we also ran mixed-effect models and compared the results using likelihood ratio testing. The model incorporated the sampling weight into newborn/perinatal deaths, sociodemographic characteristics, and health- and obstetric-related characteristics based on the probability of women being selected in the sample. We weighted respondents' data to take the number of sampled women, including those who were selected but refused to respond, by increasing the weight of the respondents. We ran separate regression models for the mean, median, and maximum quality scores as explained in the previous section.

We analyzed data from all eligible women and sub-samples of the facility and non-facility delivery cases separately. We included non-facility delivery cases since women may have received other care at a health facility, and it might have influenced the outcomes. For instance, women might have received ANC, but have not received delivery care; women might have sought care at a health facility right after delivery at home owing to an unexpected life-threatening event that occurred during childbirth. We calculated the odds ratio (OR) to examine the sign and size of the association between outcome and exposure variables. We found no missing values in the variables we used from the NDHS and NSPA. Variance inflation factors (VIF) were used to assess multicollinearity, and we set a VIF of less than four as not having considerable collinearity [39] (Table B in S1 Text). For significance testing, we set a p-value of less than .05 as significant. For all procedures, STATA version 17 (StataCorp LLC, College Station, TX, United States, 2021) was used to analyze the data.

### 2.6. Inclusivity in global research

Additional information regarding the ethical, cultural, and scientific considerations specific to inclusivity in global research is included in S1 Checklist.

## 3. Results

### 3.1. Characteristics of the delivery facilities and women

Table 1 shows the characteristics of the delivery facilities in NSPA 2015. Of the 623 surveyed facilities, 38.0 percent were health posts (n = 237), and 20.1 percent were in province 3 (n = 125), where Kathmandu was located. The median number of maternity beds per health facility was 2.0 (interquartile range [IQR] = 1.0–4.0), and the median number of SBA (physician, nurse, and auxiliary nurse midwife) was 4.0 (IQR = 3.0–10.0).

Table 2 shows the characteristics of the women in the ANC exit interviews in NSPA 2015 (n = 1,509). Of the total, 27.5 percent used primary healthcare centers. The mean age of the women was 23.4 years (standard deviation [SD] 4.3), 33.7 percent belonged to Brahmin/Chhetri ethnicity, and most had a secondary (44.0 percent) or higher education (24.5 percent).

Table 3 shows the characteristics of the women who had given birth within five years of the interview in NDHS 2016 (n = 5,060). In this population, the neonatal mortality rate was 20 per 1,000 live births, the perinatal mortality rate was 29 per 1,000 live births, and the facility

**Table 1. Characteristics of facilities providing normal delivery care (n = 623).**

| Variables | n | % |
|---|---|---|
| Facility type | | |
| Government hospital | 76 | 12.2 |
| Private hospital | 96 | 15.4 |
| NGO/private (not-for-profit)/mission-based health facility | 22 | 3.5 |
| Primary healthcare center | 192 | 30.8 |
| Health post | 237 | 38.0 |
| Providing cesarean section | | |
| Yes | 136 | 21.8 |
| No | 487 | 78.2 |
| Number of maternity beds per health facility | | |
| Median = 2.0, IQR = 1.0–4.0 (excluding don't know: n = 4) | | |
| SBA (Physician, nurse, auxiliary nurse midwife) per health facility | | |
| Median = 4.0, IQR = 3.0–10.0 | | |
| Province | | |
| Province 1: Koshi | 117 | 18.8 |
| Province 2: Madhesh | 67 | 10.8 |
| Province 3: Bagmati | 125 | 20.1 |
| Province 4: Gandaki | 69 | 11.1 |
| Province 5: Lumbini | 97 | 15.6 |
| Province 6: Karnali | 61 | 9.8 |
| Province 7: Sudurpashchim | 87 | 14.0 |

**Table 2. Characteristics of women in ANC exit interview in NSPA 2015 (n = 1,509).**

| Variables | n | % |
|---|---|---|
| Facility type | | |
| Government hospital | 371 | 24.6 |
| Private hospital | 266 | 17.6 |
| NGO/private (not-for-profit)/mission-based health facility | 151 | 10.0 |
| Primary healthcare center | 415 | 27.5 |
| Health post | 306 | 20.3 |
| Age (Mean = 23.4, SD 4.28) | | |
| ≤ 19 | 258 | 17.1 |
| 20–24 | 700 | 46.4 |
| 25–29 | 393 | 26.0 |
| 30–34 | 114 | 7.6 |
| 35–45 | 33 | 2.2 |
| Don't know | 11 | 0.7 |
| Ethnicity | | |
| Brahmin/Chhetri | 509 | 33.7 |
| Other terai caste | 281 | 18.6 |
| Dalit | 194 | 12.9 |
| Janajati/Newar | 441 | 29.2 |
| Muslim/other | 84 | 5.6 |
| Education | | |
| No education | 311 | 20.6 |
| Primary | 165 | 10.9 |
| Secondary | 664 | 44.0 |
| Higher | 369 | 24.5 |

**Table 3. Characteristics of women who gave birth in the last 5 years in NDHS 2016 (n = 5,060).**

| Variables | All cases (weighted) | |
|---|---|---|
| | n | % |
| Experienced neonatal death last 5 years | | |
| No | 4957 | 98.0 |
| Yes | 103 | 2.0 |
| Experienced perinatal death last 5 years | | |
| No | 4915 | 97.1 |
| Yes | 145 | 2.9 |
| Facility delivery | 2903 | 57.4 |
| Government hospital | 1536 | 52.9 |
| Private hospital | 516 | 17.8 |
| NGO's health facility | 32 | 1.1 |
| Primary healthcare center | 140 | 4.8 |
| Health post | 508 | 17.5 |
| India | 171 | 5.9 |
| Age | Mean = 26.3, SD 0.1 | |
| ≤ 19 | 391 | 7.7 |
| 20–24 | 1665 | 32.9 |
| 25–29 | 1780 | 35.2 |
| 30–34 | 806 | 15.9 |
| 35 ≤ | 419 | 8.3 |
| Ethnicity | | |
| Brahmin/Chhetri | 1396 | 27.6 |
| Other terai caste | 1021 | 20.2 |
| Dalit | 695 | 13.7 |
| Janajati/Newar | 1573 | 31.1 |
| Muslim/Other | 375 | 7.4 |
| Education | | |
| No education | 1733 | 34.2 |
| Primary | 1019 | 20.1 |
| Secondary | 1617 | 32.0 |
| Higher | 691 | 13.7 |
| Residence | | |
| Rural | 2330 | 46.0 |
| Urban | 2730 | 54.0 |
| Wealth index | | |
| Poorest | 1082 | 21.4 |
| Poorer | 1072 | 21.2 |
| Middle | 1121 | 22.2 |
| Richer | 1036 | 20.5 |
| Richest | 748 | 14.8 |
| Intended pregnancy | | |
| Later/no more | 905 | 17.9 |
| Then | 4155 | 82.1 |
| Multipara | | |
| No | 1498 | 29.6 |
| Yes | 3562 | 70.4 |
| Involvement in decision-making on health | | |

*(Continued)*

**Table 3.** (Continued)

| Variables | All cases (weighted) | |
|---|---|---|
| | **n** | **%** |
| No | 2583 | 51.0 |
| Yes | 2477 | 49.0 |
| Multiple pregnancies | | |
| No | 5033 | 99.5 |
| Yes | 27 | 0.5 |
| Tobacco (including chewing tobacco) | | |
| No | 4771 | 94.3 |
| Yes | 289 | 5.7 |

delivery rate was 57.4 percent. Most of the women used government hospitals (52.9 percent), followed by private hospitals (17.8 percent) and health posts (17.5 percent). The mean age of the women was 26.3 years (SD 0.1).

## 3.2. Comparison of additive indicators and PCA methods

All three dimensions had high correlation coefficients between the quality scores derived from the two methods (Dimension 1: r = 0.99, Dimension 2: r = 0.74, Dimension 3: r = 1.00; Table C in S1 Text). We used quality scores derived from the additive indicator method in the main analysis. The mean scores of the 16 quality indicators are in Fig C in S1 Text.

## 3.3. Association between quality scores and newborn deaths

Table 4 shows the results of the regression analysis examining the association between district-level maternal and newborn care quality and newborn deaths. In this section, we describe the results of the analysis using the mean and maximum quality scores in the three dimensions, which were calculated from SPA. We also describe the results of the analysis using all cases and sub-sample analyses of facility delivery cases and non-facility delivery cases. These sub-samples were divided based on DHS. The adjusted model incorporated women's characteristics from DHS as covariates. For all dimensions, the signs and sizes of the associations were largely consistent between the models using the mean and median quality scores (Table D in S1 Text). We run the mixed-effect logistic regression (using Stata's melogit command) that considered the existence of the cluster effects, in addition to the logistic regression. After running both regression models, we tested the existence of the cluster effects using the likelihood-ratio test (using Stata's lrtest command). Since the test did not support the existence of the cluster effect, we used the result of logistic regression models.

   **3.3.1. Dimension 1: Provision of care.**   In the analysis of all delivery cases, a higher mean Quality Score 1 (provision of care) was significantly associated with a lower number of newborn deaths (Adjusted odds ratio [AOR] = 0.04, 95 percent confidence interval [95% CI]: 0.00–0.76). In the sub-sample analysis of facility delivery cases only, a higher mean Quality Score 1 was significantly associated with a lower number of newborn deaths (AOR = 0.01, 95% CI: 0.00–0.86). On the other hand, in the sub-sample analysis of non-facility delivery cases only, the mean Quality Score 1 was not significantly associated with newborn deaths (AOR = 0.18, 95% CI: 0.01–4.81).

   In the analysis of all delivery cases, a higher maximum Quality Score 1 was associated with a lower number of newborn deaths (AOR = 0.09, 95% CI: 0.01–0.58). In the sub-sample analysis of facility delivery cases only, the maximum Quality Score 1 was not significantly associated

**Table 4. Binary logistic regression: Association between the quality scores and newborn deaths using mean and max scores.**

| Quality dimension | Samples(weighted) | Analysis using MEAN score | | | | Analysis using MAX score | | | |
|---|---|---|---|---|---|---|---|---|---|
| | | OR (95% CI) | p | AOR* (95% CI) | p | OR (95% CI) | P | AOR* (95% CI) | p |
| **Quality Score 1: Provision of care** | **a. All cases** | 0.05 | < .01 | 0.04 | **.03** | 0.11 | **.01** | 0.09 | **.01** |
| | **(n = 5,060)** | (0.01–0.46) | | (0.00–0.76) | | (0.02–0.60) | | (0.01–0.58) | |
| | **b. Facility delivery cases only** | 0.02 | **.01** | 0.01 | **.04** | 0.11 | .11 | 0.14 | .29 |
| | **(n = 2,903)** | (0.00–0.40) | | (0.00–0.86) | | (0.01–1.68) | | (0.00–5.27) | |
| | **c. Non-facility delivery cases only** | 0.18 | .25 | 0.18 | .31 | 0.17 | .10 | 0.09 | .05 |
| | **(n = 2,157)** | (0.01–3.50) | | (0.01–4.81) | | (0.02–1.39) | | (0.01–1.02) | |
| **Quality Score 2: Experience of care** | **a. All cases** | 0.36 | .44 | 0.61 | .73 | 0.37 | .22 | 0.57 | .47 |
| | **(n = 5,060)** | (0.03–4.78) | | (0.04–10.39) | | (0.08–1.81) | | (0.12–2.66) | |
| | **b. Facility delivery cases only** | 0.03 | .11 | 0.06 | .26 | 0.63 | .69 | 1.02 | .98 |
| | **(n = 2,903)** | (0.00–2.09) | | (0.00–7.91) | | (0.06–6.23) | | (0.09–11.65) | |
| | **c. Non-facility delivery cases only** | 3.43 | .46 | 3.97 | .47 | 0.33 | .33 | 0.45 | .44 |
| | **(n = 2,157)** | (0.13–90.68) | | (0.09–174.14) | | (0.04–3.09) | | (0.06–3.50) | |
| **Quality Score 3: Human and physical resources** | **a. All cases** | 0.32 | .18 | 2.96 | .35 | 0.62 | .69 | 1.70 | .69 |
| | **(n = 5,060)** | (0.06–1.70) | | (0.30–29.00) | | (0.06–6.46) | | (0.12–23.22) | |
| | **b. Facility delivery cases only** | 0.19 | .21 | 9.97 | .28 | 0.39 | .66 | 1.85 | .82 |
| | **(n = 2,903)** | (0.01–2.54) | | (0.16–637.47) | | (0.01–26.19) | | (0.01–339.69) | |
| | **c. Non-facility delivery cases only** | 0.83 | .87 | 1.46 | .79 | 1.27 | .87 | 2.44 | .54 |
| | **(n = 2,157)** | (0.08–8.17) | | (0.08–25.02) | | (0.08–19.31) | | (0.14–42.72) | |

OR = Odds ratio, AOR = Adjusted odds ratio, CI = Confidence interval

* Adjusted for age, ethnicity, education, residential area, wealth index, intended pregnancy, multipara, involved in decision making on health, multiple pregnancies, tobacco and province.

with newborn deaths (AOR = 0.14, 95% CI: 0.00–5.27). In the sub-sample analysis of non-facility delivery cases only, the maximum Quality Score 1 was marginally, but not significantly, associated with newborn deaths (AOR = 0.09, 95% CI: 0.01–1.02).

**3.3.2. Dimension 2: Experience of care.** The mean Quality Score 2 (experience of care) had no significant association with newborn deaths (all cases: AOR = 0.61, 95% CI: 0.04–10.39). The maximum Quality Score 2 had no significant association with newborn deaths (all cases: AOR = 0.57, 95% CI: 0.12–2.66).

**3.3.3. Dimension 3: Human and physical resources.** The mean Quality Score 3 (human and physical resources) had no significant association with newborn deaths (all cases: AOR = 2.96, 95% CI: 0.30–29.00). A maximum Quality Score 3 had no significant association with newborn deaths (all cases: AOR = 1.70, 95% CI: 0.12–23.22).

## 3.4. Association between quality scores and perinatal deaths

Table 5 shows the results of the regression analysis that examined the association between district-level maternal and newborn care quality and perinatal deaths. For all dimensions, the signs and sizes of the associations were largely consistent between the models using the mean and median quality scores (Table D in S1 Text). In all cases and sub-samples, the mean quality scores for all three dimensions were not significantly associated with perinatal deaths.

**Table 5. Binary logistic regression: Association between the quality scores and perinatal deaths using mean and max scores.**

| Quality dimension | Samples (weighted) | Analysis using MEAN score | | | | Analysis using MAX score | | | |
|---|---|---|---|---|---|---|---|---|---|
| | | OR (95% CI) | p | AOR* (95% CI) | p | OR (95% CI) | p | AOR* (95% CI) | p |
| **Quality score 1: Provision of care** | **a. All cases** | 0.24 | .15 | 0.32 | .37 | 0.18 | **.03** | 0.23 | .10 |
| | (n = 5,060) | (0.03–1.69) | | (0.03–3.86) | | (0.04–0.88) | | (0.04–1.31) | |
| | **b. Facility delivery cases only** | 0.65 | .76 | 1.61 | .81 | 0.88 | .93 | 1.75 | .76 |
| | (n = 2,903) | (0.04–10.31) | | (0.03–76.95) | | (0.05–15.27) | | (0.05–59.43) | |
| | **c. Non-facility delivery cases only** | 0.21 | .20 | 0.13 | .20 | 0.14 | **.02** | 0.15 | .07 |
| | (n = 2,157)** | (0.02–2.31) | | (0.01–2.90) | | (0.03–0.73) | | (0.02–1.17) | |
| **Quality score 2: Experience of care** | **a. All cases** | 0.27 | .30 | 0.92 | .95 | 0.41 | .23 | 0.56 | .48 |
| | (n = 5,060) | (0.02–3.32) | | (0.07–11.84) | | (0.10–1.74) | | (0.12–2.73) | |
| | **b. Facility delivery cases only** | 0.04 | .13 | 0.09 | .24 | 1.05 | .97 | 1.77 | .68 |
| | (n = 2,903) | (0.00–2.47) | | (0.00–5.19) | | (0.11–10.00) | | (0.11–28.19) | |
| | **c. Non-facility delivery cases only** | 1.48 | .79 | 8.38 | .23 | 0.30 | .19 | 0.27 | .19 |
| | (n = 2,157)** | (0.08–27.25) | | (0.25–276.67) | | (0.05–1.82) | | (0.04–1.93) | |
| **Quality score 3: Human and physical resources** | **a. All cases** | 0.33 | .15 | 0.92 | .95 | 0.66 | .70 | 1.16 | .91 |
| | (n = 5,060) | (0.07–1.51) | | (0.07–11.84) | | (0.08–5.40) | | (0.10–13.25) | |
| | **b. Facility delivery cases only** | 0.22 | .17 | 3.60 | .43 | 1.12 | .96 | 4.21 | .62 |
| | (n = 2,903) | (0.03–1.88) | | (0.15–87.47) | | (0.01–112.79) | | (0.02–1158.03) | |
| | **c. Non-facility delivery cases only** | 0.85 | .88 | 0.95 | .97 | 0.97 | .98 | 1.11 | .94 |
| | (n = 2,157)** | (0.10–6.96) | | (0.06–14.98) | | (0.10–9.60) | | (0.07–17.56) | |

OR = Odds ratio, AOR = Adjusted odds ratio, CI = Confidence interval

* Adjusted for age, ethnicity, education, residential area, wealth index, intended pregnancy, multipara, involved in decision making on health, tobacco and province.

** Omitted multiple pregnancies. Other covariates are the same above.

In the analysis of all delivery cases, the maximum quality scores in all three dimensions were not significantly associated with perinatal deaths in the adjusted model, although Quality Score 1 was significantly associated with a lower number of perinatal deaths in the non-adjusted model (OR = 0.18, 95% CI: 0.04–0.88). In the sub-sample analysis of facility delivery cases only, the maximum quality score in all dimensions was not significantly associated with perinatal deaths. In the sub-sample analysis of non-facility delivery cases only, the maximum Quality Score 1 was significantly associated with perinatal deaths in the non-adjusted model, but not in the adjusted model (OR = 0.14, 95% CI: 0.03–0.73; AOR = 0.15, 95% CI: 0.02–1.17). Quality Scores 2 and 3 were not associated with perinatal deaths.

## 4. Discussion

This study systematically examined the association between the level of care quality and newborn and perinatal deaths in Nepal in three dimensions. By linking two nationally representative datasets from health facilities (NSPA) and women (NDHS), we investigated the district-level quality of the health system. The higher quality scores in care provision at its average and highest levels in each district were associated with a lower number of newborn deaths, but not with perinatal deaths. Moreover, the quality of care experience and human and physical resources were not associated with either newborn or perinatal death.

### 4.1. Association between quality of care and newborn deaths

In this study, both higher mean and maximum quality scores in the provision of care were significantly associated with a lower number of newborn deaths. As high-quality facilities provide high-quality BEmOC and information systems, these factors may have contributed to the reduction in newborn deaths. In particular, BEmOC is known to reduce intrapartum-related newborn deaths by 40 percent, whereas delivery care by SBA without access to the emergency component reduces them by 25 percent [32, 33]. Providing high-quality care is crucial, as facility delivery alone does not always result in improved newborn health outcomes [2, 40]. Although the higher quality in the provision of care might contribute to reducing newborn deaths, no significant association was detected for non-facility delivery.

The analysis using the maximum quality score in non-facility delivery cases only showed marginally, but not a significant result. In this analysis, the adjusted odds ratio could fall between 0.01 and 1.02. Since this was a sub-sample analysis with a smaller sample size, they might have a significant association with a larger sample size. In contrast, the analysis of facility delivery cases did not have a significant association. In a study in India, newborns who came to a referral hospital directly from home had a higher survival rate than those who came from other lower-tier health facilities [41]. In this study, the facility with the highest quality score in the district might have saved newborns' lives, such as whose mother had received ANC but had not received delivery care; whose mother had sought care at a health facility right after delivery owing to an unexpected life-threatening event occurred during childbirth. Enhancing the quality of care in Dimension 1 in at least one health facility in each district may be a potential step toward reducing newborn deaths. Based on the results of this study, it would be a plausible strategy to invest in quality improvement for the highest-tier health facility in a district, particularly in settings resource-constrained and with low facility delivery coverage. No significant association was detected between the quality scores in the experience of care and newborn deaths. To measure the experience of care, the following items were included: "effective communication," "respect and preservation of dignity," and "emotional support." These items are components of person-centered maternity care (PCMC). In this sense, the results of this study are consistent with the results of a previous study that showed that PCMC had no clear association with both clinical processes and clinical outcomes [42]. However, PCMC is known to have an impact on the clinical process of care, such as decreased cesarean section, more intact perineum, and shorter labor time in high-income settings [42]. However, these outcomes were not measured in this study.

No significant association was detected between the quality scores in human and physical resources and newborn deaths. This is possibly because this study used data on resource availability, but not functionality. For instance, this study did not reflect the water quality, convenience of water source location, and providers' adherence to hand washing, even if a facility had access to an improved water source. In a study that assessed the quality of maternal and newborn health in eight LMICs, the correlation between inputs at health facilities (e.g., amenities, equipment, and medications) and clinical care quality was weak [15]. Functional factors might be necessary to detect the association between this quality score and newborn deaths.

### 4.2. Association between quality of care and perinatal deaths

No significant association was detected between the three quality scores and perinatal death. For the provision of care, this study mainly focused on the quality of delivery care; however, the quality of ANC, family planning, and comprehensive emergency obstetric care might also be associated with perinatal deaths. In southern Asia, 39.7 percent of stillbirths occurred during the antenatal period, and they could have been prevented with high-quality ANC [30].

More specifically, the population-attributable risk of stillbirth is the highest in lasting pregnancies longer than 42 weeks, followed by pre-existing hypertension and diabetes and maternal infection by syphilis in southern Asia [36]. They are preventable with sufficient care during the antenatal period, such as diagnosis of exact gestational age and preventing and controlling infection and non-communicable disorders. In a study from Zimbabwe, improvement in the quality of ANC was associated with decreased neonatal mortality [43]. Moreover, a timely cesarean section can contribute to reducing intrapartum stillbirth [36, 44]. The quality of these ANC and advanced obstetric care, which this study did not include owing to the limited availability of data, might have a stronger association with perinatal deaths. Also, the first and second delay in the three-delay model, which occurs before reaching health facilities, may have contributed more when it comes to the perinatal health outcome. Patient-oriented factors and transport from home to facilities may take longer time before labor started [45].

### 4.3. Strengths and limitations

This study had three strengths. First, this study examined maternal and newborn care quality and newborn/perinatal deaths using nationally representative datasets. Evidence on the association between care quality and newborn/perinatal deaths is limited, mostly owing to the lack of necessary data for analysis. By using two datasets at the national level, this study provided evidence of the association between the level of care quality and newborn and perinatal deaths. Second, this study investigated the quality of district-level care by linking two nationally representative surveys. Quality of care is often described at the individual or facility level. However, this study described district-level quality and covered the neglected part of the quality of the health system, such as interactions among health providers and gaps between clinical settings and health policy. Third, this study systematically examined detailed multidimensional qualities, including women's experiences, in the context of Nepal. Moreover, this study reflected women's characteristics in the community by linking the two surveys. The results of this study can be applied to other settings in low- and lower-middle-income countries to improve maternal and newborn health.

However, this study had some limitations. First, although SPA incorporated a comprehensive assessment including facility readiness, clinical care, and user experience, the available indicators may not have comprehensively covered care quality. Dimension 1 did not include fetal and newborn indicators (e.g., fetal heart rate monitoring), and Dimension 2 included indicators related to ANC but not delivery care owing to the limited availability of data. However, by using data from SPA, we used more indicators of user experience compared with other global and cross-national quality measure sets [1, 46]. Second, quality scores in the provision of care and human and physical resources might have less reflected in health facilities with more delivery volume. The quality scores were unweighted, and all health facilities were treated equally, regardless of delivery volume. If the delivery volume at each facility were available, this study could have reflected it in adjusting the representativeness of the health facilities. Third, we did not include the district-level factors which may influence the health facility's performance, such as local epidemiology and local health management. We used basic and essential indicators and limited the role of such factors [47]. Further investigation is required at the district level, including local context, governance, and funding [8]. Lastly, we might have connected individual women with different facilities which they utilized when we linked data at the district level. For example, women who sought care for delivery in another district might have been misconnected since they did not meet the assumption that they used a facility in their residential district. Nonetheless, we could eliminate the mismatch owing to cluster displacement in the DHS sampling procedure and women's preference for health facilities within

the district by using the administrative boundary methods. This method also enabled the assessment of not individual facility quality, but a district-level health system. We examined the association between the quality of care and health outcomes using the representative of care quality of the district by using this method even though it was not a census of health facilities.

## 5. Conclusions

This study showed that district-level quality improvement may reduce newborn deaths, but not perinatal deaths, by enhancing the quality of care provision at its average and highest levels in each district. This study showed that quality of care experience and human and physical resources had no significant association with newborn and perinatal deaths. Health administrators should assess the quality of care at the administrative division level so that they can have a high-quality health facility and improve the average level of care quality in health facilities in the division. The government of Nepal has already recommended expanding the provision of BEmOC, which we used as an indicator of the provision of care [48]. This study adds evidence to support this goal. Nepal requires further accelerations to enhance the quality of care provision. This finding can be applied to other low- and lower-middle-income countries.

This study was a secondary data analysis, which verified that individual data from women and newborns and health facility data were linked to assess the quality of care that women and newborns could potentially receive. This study provided hypotheses for future research that enhances what care quality dimensions could result in improving newborn and perinatal health outcomes in the limited availability of data. To further verify the quality of care that women and newborns receive, further studies are needed with rigorous methods that can support the findings of our study. Also, the datasets can be modified to promote analysis using data linkage methods by incorporating the variables which could easily link datasets (e.g., health facilities that women used), and narrowing the time difference between surveys. This study has two implications for future research. First, future studies should identify quality components that would improve perinatal health outcomes, such as the quality of ANC, advanced maternal and newborn care, and family planning. Second, future studies should investigate outcome-specific measurements for the quality of care. Resource readiness, which was widely used to measure the care quality, may not be associated with improving health outcomes. Since the concept of quality of care is broad, researchers should correctly choose which indicators to incorporate depending on the health outcomes to achieve when measuring the quality of care.

## Supporting information

**S1 Checklist. Inclusivity in global research.**
(DOCX)

**S2 Checklist. STROBE statement.**
(DOCX)

**S1 Text.**
(DOCX)

## Acknowledgments

We are grateful to the DHS program for allowing us to use the NSPA and NDHS dataset for this study. We would like to thank Dr. Sumiyo Okawa for her expert commenting during

proposal development. We would like to thank Editage (www.editage.com) for English language editing.

## Author Contributions

**Conceptualization:** Subaru Ikeda, Akira Shibanuma.

**Data curation:** Subaru Ikeda.

**Formal analysis:** Subaru Ikeda, Akira Shibanuma.

**Funding acquisition:** Subaru Ikeda, Akira Shibanuma.

**Investigation:** Subaru Ikeda, Akira Shibanuma, Alpha Pokharel, Ram Chandra Silwal, Masamine Jimba.

**Methodology:** Subaru Ikeda.

**Project administration:** Subaru Ikeda, Akira Shibanuma.

**Resources:** Subaru Ikeda.

**Software:** Subaru Ikeda.

**Supervision:** Subaru Ikeda, Masamine Jimba.

**Validation:** Subaru Ikeda.

**Visualization:** Subaru Ikeda.

**Writing – original draft:** Subaru Ikeda, Alpha Pokharel.

**Writing – review & editing:** Akira Shibanuma, Ram Chandra Silwal, Masamine Jimba.

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
