## [Decision Letter · Decision Letter 0]

19 May 2023

PGPH-D-23-00568

Improving the quality of maternal and newborn healthcare at the district level: addressing newborn deaths in Nepal

Dear Dr. Akira

Thank you for submitting your manuscript to PLOS Global Public Health. After careful consideration, we feel that it has merit but does not fully meet PLOS Global Public Health’s publication criteria as it currently stands. Therefore, we invite you to submit a revised version of the manuscript that addresses the points raised during the review process.

The paper -reviewers have shared a positive outcome with a few suggestions. Please review these and address them.

The paper would benefit from a careful re-read and a finer editorial check for typos, and inconsistencies, especially in the tables. I am sharing a few additional points where errors were noted.I would encourage the authors to address this as best as they can. Re-reviewed and incomplete corrections are not encouraged as they add to the review process and are unfavourably viewed.

Style consistency for numbers under 10 spell them out

Please use per cent instead of % in the text

115-117 please qualify this line  in the case of Nepal.The aspects mentioned in line 116 have been studied elsewhere with known care outcomes across the maternal-newborn spectrum.

Again in line 126 please qualify that the existent study gap is for Nepal.Care outcomes have been studied at the district level for South Asian countries.

Lines 135Check intext referencing format and follow journal guidelines.Concurrent references are separated by a dash or comma, eg [21][22] should be written as [21,22].

Please explain in line 139 how the 992 health facilities were selected from the larger pool of health facilities. Do these include both public and private care facilities?

Table 1: please list province names

Please check table 2 formatting (NGO/private… shows % typos’’

We look forward to receiving your revised manuscript.

Kind regards,

Danish Ahmad, MBBS,MSc,MNAMS,PhD

Academic Editor

Journal Requirements:

Additional Editor Comments (if provided):

The paper -reviewers have shared a positive outcome with a few suggestions. Please review these and address them.

The paper would benefit from a careful re-read and a finer editorial check for typos, and inconsistencies, especially in the tables. I am sharing a few additional points where errors were noted.I would encourage the authors to address this as best as they can. Re-reviewed and incomplete corrections are not encouraged as they add to the review process and are unfavourably viewed.

Style consistency for numbers under 10 spell them out

Please use per cent instead of % in the text

115-117 please qualify this line in the case of Nepal.The aspects mentioned in line 116 have been studied elsewhere with known care outcomes across the maternal-newborn spectrum.

Again in line 126 please qualify that the existent study gap is for Nepal.Care outcomes have been studied at the district level for South Asian countries.

Lines 135Check intext referencing format and follow journal guidelines.Concurrent references are separated by a dash or comma, eg [21][22] should be written as [21,22].

Please explain in line 139 how the 992 health facilities were selected from the larger pool of health facilities. Do these include both public and private care facilities?

Table 1: please list province names

Please check table 2 formatting (NGO/private… shows % typos’’

Reviewers' comments:

Reviewer's Responses to Questions

**Comments to the Author**

1. Does this manuscript meet PLOS Global Public Health’s publication criteria? Is the manuscript technically sound, and do the data support the conclusions? The manuscript must describe methodologically and ethically rigorous research with conclusions that are appropriately drawn based on the data presented.

Reviewer #1: Yes

Reviewer #2: Yes

2. Has the statistical analysis been performed appropriately and rigorously?

Reviewer #1: Yes

Reviewer #2: Yes

3. Have the authors made all data underlying the findings in their manuscript fully available (please refer to the Data Availability Statement at the start of the manuscript PDF file)?

Reviewer #1: Yes

Reviewer #2: Yes

4. Is the manuscript presented in an intelligible fashion and written in standard English?

Reviewer #1: Yes

Reviewer #2: Yes

5. Review Comments to the Author

Reviewer #1: Although the analysis is rigorous and the data available in 2015 and 2016 have been used well for the purpose of the study, Newborn care particularly following newer studies related to skin-to-skin care and Kangaroo Mother Care now propose different approaches to Newborn care quality that were not known well during the surveys conducted in 2015 and 2016. Further later studies such as the SPA 2021 in Nepal provide richer information. It may be useful for the authors to note some of these advances and publications and how the quality of care is undergoing major shifts when it comes to newborn care and preterm babies. If not, the reader in 2023 may still use the parameters measured and surveyed in 2015 and 2016 for decisions related to how one can make a major shift to reducing newborn mortality.

When it comes to Perinatal mortality issues of 3 delays related to complications occurring during pregnancy and how they were handled at the family level, at the time of transport from home to the facility and at the facility itself are important. Perhaps these can be included in the discussion even if there were no data. Maternal mortality and morbidity and the main causes of complications can occur regardless of the ANC quality and the authors could note these in their discussion.

Reviewer #2: The topic is relevant to Nepal and in resource limited settings.

The narration was intelligently presented

Analysis plan was detailed and Comprehensive

Results and discussion are in line with the research questions

6. PLOS authors have the option to publish the peer review history of their article (what does this mean?). If published, this will include your full peer review and any attached files.

**Do you want your identity to be public for this peer review?** For information about this choice, including consent withdrawal, please see our Privacy Policy.

Reviewer #1: **Yes: **Lakshmi Narasimhan Balaji

Reviewer #2: No

---

## [Editor Report · Decision Letter 1]

29 Jun 2023

PGPH-D-23-00568R1

Improving the quality of maternal and newborn healthcare at the district level: addressing newborn deaths in Nepal

Dear Dr. Shibanuma 

Thank you for submitting your manuscript to PLOS Global Public Health. After careful consideration, we feel that it has merit but does not fully meet PLOS Global Public Health’s publication criteria as it currently stands. Therefore, we invite you to submit a revised version of the manuscript that addresses the points raised during the review process.

We look forward to receiving your revised manuscript.

Kind regards,

Danish Ahmad, MBBS,MSc,MNAMS,PhD,FRCP

Academic Editor

Journal Requirements:

2. Please insert an Ethics Statement at the beginning of your Methods section, under a subheading 'Ethics Statement'. It must include:

1) The name(s) of the Institutional Review Board(s) or Ethics Committee(s)

2) The approval number(s), or a statement that approval was granted by the named board(s) 

3) (for human participants/donors) - A statement that formal consent was obtained (must state whether verbal/written) OR the reason consent was not obtained (e.g. anonymity). NOTE: If child participants, the statement must declare that formal consent was obtained from the parent/guardian.

Additional Editor Comments (if provided):

Thank you for revising the paper-there are two un addressed comments that the authors should address

Please correct the typo in these lines-"Although a study in another South Asian country 123 examined the geographic variation of maternal and newborn care quality at the district level in 124 another south Asian country, it also focused on inputs of routine care but not multidimensional

125 care quality [19].

2. referring to comment 7 where province names were requested to be listed, the NDHS 2016 report on page 13 provides a map with corresponding province numbers-can you pleae try to add provinvce names by reviewing the map and matching against exisitn names. Publishing a paper with province numbers and not numbers limitts interpretation for international and importantly researchers/policy and program managers in Nepal
---

## [Editor Report · Decision Letter 2]

11 Jul 2023

Improving the quality of maternal and newborn healthcare at the district level: addressing newborn deaths in Nepal

PGPH-D-23-00568R2

Dear Dr Akira

We are pleased to inform you that your manuscript 'Improving the quality of maternal and newborn healthcare at the district level: addressing newborn deaths in Nepal' has been provisionally accepted for publication in PLOS Global Public Health.

Best regards,

Danish Ahmad, MBBS,MSc,MNAMS,PhD,FRCP

Academic Editor

Thank you for addressing the pending queries, I am pleased to recommend the paper for publication.